# Highly multiplexed quantitative PCR-based platform for evaluation of chicken immune responses

Dominika Borowska[1]*, Richard Kuo[1], Richard A. Bailey[2], Kellie A. Watson[1,2], Pete Kaiser[1†], Lonneke Vervelde[1], Mark P. Stevens[1]

**1** The Roslin Institute and Royal (Dick) School of Veterinary Studies, University of Edinburgh, Easter Bush, Midlothian, Scotland, United Kingdom, **2** Aviagen Ltd, Edinburgh, Scotland, United Kingdom

† Deceased.
* dominika.borowska@roslin.ed.ac.uk

**Data Availability Statement:** The RNA-seq dataset generated and analysed during the current study is available in the European Bioinformatics Institute ArrayExpress using accession number E-MTAB-

## Abstract

To address the need for sensitive high-throughput assays to analyse avian innate and adaptive immune responses, we developed and validated a highly multiplexed qPCR 96.96 Fluidigm Dynamic Array to analyse the transcription of chicken immune-related genes. This microfluidic system permits the simultaneous analysis of expression of 96 transcripts in 96 samples in 6 nanolitre reactions and the 9,216 reactions are ready for interpretation immediately. A panel of 89 genes was selected from an RNA-seq analysis of the transcriptional response of chicken macrophages, dendritic cells and heterophils to agonists of innate immunity and from published transcriptome data. Assays were confirmed to be highly specific by amplicon sequencing and melting curve analysis and the reverse transcription and preamplification steps were optimised. The array was applied to RNA of various tissues from a commercial line of broiler chickens housed at two different levels of biosecurity. Gut-associated lymphoid tissues, bursa, spleen and peripheral blood leukocytes were isolated and transcript levels for immune-related genes were defined. The results identified blood cells as a potentially reliable indicator of immune responses among all the tissues tested with the highest number of genes significantly differentially transcribed between birds housed under varying biosecurity levels. Conventional qPCR analysis of three differentially transcribed genes confirmed the results from the multiplex qPCR array. A highly multiplexed qPCR-based platform for evaluation of chicken immune responses has been optimised and validated using samples from commercial chickens. Apart from applications in selective breeding programmes, the array could be used to analyse the complex interplay between the avian immune system and pathogens by including pathogen-specific probes, to screen vaccine responses, and as a predictive tool for immune robustness.

2996 https://www.ebi.ac.uk/arrayexpress/experiments/E-MTAB-2996/.

**Funding:** MPS and PK received funding for a doctoral studentship for DB from the Biotechnology and Biological Sciences Research Council (BBSRC) and Aviagen Ltd (reference BB/J500744/1). This funded a stipend for DB and part of the directly-incurred research costs of the project. In addition, Aviagen Ltd. provided in-kind contributions, including access to samples from the broiler populations under study and the input of its employees. KAW was employed by Aviagen Ltd. during part of the study and RAB was employed by Aviagen Ltd throughout and to date. MPS and LV received funding for part of their salaries and research costs from BBSRC Institute Strategic Programmes Grants to The Roslin Institute (references BBS/E/D/10002071 and BBS/E/D/20002174); MPS and LV also received funding from the European Union Horizon 2020 research and innovation programme under grant agreement N°731014 (VetBioNet), which supported part of the salary of DB and directly-incurred costs of the research described. Beyond the contributions described above, the funders did not have any additional role in the study design, data collection and analysis, decision to publish, or preparation of the manuscript. The specific roles of the authors are articulated in the 'author contributions' section.

**Competing interests:** We declare that the study received financial and in-kind support from Aviagen Ltd. This included a contribution to the stipend of DB while a doctoral candidate at The University of Edinburgh and in-kind contributions including access to broiler populations and the provision of biological materials. For part of the study KAW was an employee of Aviagen Ltd. and RAB has been employed by the company throughout and to date. This does not alter our adherence to PLOS ONE policies on sharing data and materials. Provision of biological materials derived from Aviagen broilers may be subject to a Material Transfer Agreement covering their secondary uses.

## Introduction

Poultry are vital to global food security and the Food and Agriculture Organisation of the United Nations estimated that 75 billion broilers and 1.9 trillion eggs were produced in 2016. Populations of poultry, whether reared in intensive commercial operations or extrinsic scavenging systems, are exposed to diverse viral, prokaryotic and eukaryotic pathogens that can produce high morbidity and/or mortality. Vaccines have helped to control avian and zoonotic diseases, but are not available for many pathogens and an urgent need exists to better understand the basis of natural and vaccine-mediated immunity. In recent years, sequencing of avian genomes and transcriptome analyses have added to knowledge of the repertoire and function of constituents of the immune system of poultry.

Immune responses to encountered microorganisms are coordinated on cellular and molecular levels. The most exhaustive analysis of tissues and cells can be achieved by massively-parallel RNA sequencing (RNA-seq). For many years researchers focused their efforts on nucleic-acid based tools to detect transcripts, with microarrays being the most widely used tool [1]. Although the levels of mRNA inside the cells do not always correspond with the amount of proteins that will be produced, exploring immune responses at the level of transcription is more accessible, especially for farmed species owing to the paucity of antibodies for key immune cell types, receptors and ligands. Screening hundreds of targets and samples in parallel quantitative PCR (qPCR) for gene expression is possible with high-throughput qPCR tools, for example the 96.96 Dynamic Array from Fluidigm [2] or NanoString nCounter gene expression system [3]. This type of high-throughput platform would be advantageous for the rapid and cost-effective analysis of immune responses, as a diagnostic tool to screen for pathogens and as a selection tool to breed more robust chicken lines based on production traits supported by desired gene expression profiles.

While it has been possible to improve chickens by genetic selection for resistance to specific diseases, as first reported by Roberts and Card (1935) [4], achieving a general increase in immunological competence is considered challenging because of low heritability and the difficulty of measuring this trait. Selection based on immune function does not dramatically effect growth promotion, therefore it could be possible to select for immune responsiveness without causing a decline in weight gain [5]. The generation of more robust lines of birds with improved liveability can be anticipated to reduce economic losses and enhance welfare [6]. In order to do this there is a demand for a tool that could rapidly and precisely evaluate avian immune responses associated with innate immunity and disease resistance.

The objectives of this study were to develop, optimise and validate a custom multiplexed qPCR array for analysis of the transcription of immune-related genes that could be used to phenotype the immune responses of chickens at the level of innate immunity. After selection of 89 genes that are reliably differentially transcribed in response to pathogens (or constituents thereof) and 3 reference genes, qPCR assays were devised and confirmed to be highly specific and sensitive by amplicon sequencing, melting curve analysis and limiting dilution of specific transcripts. Following optimisation of reverse transcription and preamplification conditions for reactions on a nanolitre scale, it was then applied to study transcript levels in tissues of the same broiler genotype housed at two different levels of biosecurity; a high-biosecurity 'pedigree' farm and a 'sibling-test' farm that mimics the commercial farm environment [7]. The aim was to compare gene expression in gut-associated lymphoid tissues (caecal tonsils, ileum), bursa, spleen and peripheral blood leukocytes (PBLs) of related age-matched birds reared on the two farms. A further goal for this study was to compare responses in PBLs compared to internal organs as a predictor of the robustness of immune responses in the chicken, as non-lethal blood sampling would allow selective breeding from the birds screened. Although we

illustrate the application of the platform in the context of selection of birds in breeding programmes, the array may be applied to analyse the complex interplay between the avian immune system and pathogens, to screen vaccine-induced responses, and as a predictive tool for immune robustness.

## Materials and methods

### Ethical statement

Commercial Brown Leghorns J-line birds were housed in premises licensed under a UK Home Office Establishment License in full compliance with the Animals (Scientific Procedures) Act 1986 and the Code of Practice for Housing and Care of Animals Bred, Supplied or Used for Scientific Purposes. Requests for animals were approved by the Roslin Institute Animal Welfare and Ethical Review Board and animals were humanely culled in accordance with Schedule 1 of the Animals (Scientific Procedures) Act 1986.

Samples for high throughput qPCR were collected from commercial breeding farms (pedigree and sibling-test) in Scotland, UK where birds are housed and managed in line with the EU Broiler Welfare Directive. Protocols were approved and managed through the Aviagen veterinary department.

### Gene selection based on published studies

Target genes for qPCR analysis were selected partly using published data. Published datasets were identified by searching the NCBI database with the queries 'innate immune response' and 'gene expression infection' in both chicken and mammalian species, and by searches of cited references in selected articles. The publications had to contain an analysed differential expression (DE) dataset in the body of text and/or in the supplementary data available online. Infection studies on various pathogens and their interactions with the host as well as *in vitro* studies on stimulated primary cells and/or cell lines were included (S1 Table). For each article used, the differential expression (DE) dataset was compared with other selected studies and the genes that were upregulated in two or more studies were considered a candidate for the gene list.

### Sample preparation for RNA sequencing

To extend the list of genes of interest, RNA-seq analysis of the transcriptional response of chicken bone marrow-derived dendritic cells (BMDCs), bone marrow-derived macrophages (BMDMs) and blood-derived heterophils stimulated with lipopolysaccharide (LPS) was performed. The chickens used in the following experiments were Brown Leghorn-J line bred and reared in floor pens at the National Avian Research Facility, The Roslin Institute, Edinburgh (UK). The chickens were maintained under conventional conditions and received standard vaccination scheme against Marek's Disease Virus (MDV), Infectious Bursal Disease Virus (IBDV), *Eimeria spp*, Infectious Bronchitis Virus (IBV) and Newcastle Disease Virus (NDV). The chickens were housed in groups and received food and water ad libitum. All birds were considered healthy by physical examination. Two birds were humanely culled by cervical dislocation, in accordance with Schedule 1 of the Animals (Scientific Procedures) Act 1986, at six-weeks of age and bone marrow was collected. Femurs and tibias were flushed using 0.8 x 40 mm diameter needle (21G x 1.5 Terumo) and 10 ml syringe with sterile phosphate-buffered saline (PBS). Cells suspensions were passaged through 70 μm mesh strainers. Cells were pelleted at 400 x *g* for 10 min at room temperature (RT) and resuspended in 10 ml of PBS. Histopaque 1.077 was used to separate mononuclear cells by underlying the bone marrow cell

suspension and centrifugation at 400 x *g* for 20 min with the brake switched off. The interface was collected and washed with PBS at 400 x *g* for 10 min. Cells were resuspended in 10 ml complete media (RPMI-1640 medium supplemented with 10% foetal calf serum (FCS), 200 mM L-glutamine, 1U/ml of penicillin and 1 μg/ml of streptomycin) and counted. The cells were seeded in six-well plates at a concentration of 1 x $10^6$ cell/ml in a total of 3 ml of complete media supplemented with recombinant chicken interleukin 4 (IL-4) and granulocyte-macrophage colony stimulating factor (GM-CSF) (BMDCs) and colony stimulating factor (CSF-1) (BMDMs). Cells were incubated in 41˚C, 5% $CO_2$ for six days with media changed on the third day. On day six, BMDCs were stimulated with 200 ng/ml *Escherichia coli* O55:B5 lipopolysaccharide (LPS; Sigma Aldrich) for 24 h. BMDMs were stimulated with 250 ng/ml of *E. coli* LPS for 4 h. Heterophils were isolated from peripheral blood collected from one hundred day-old chickens humanely culled by cervical dislocation. Blood was collected into K2 ethylenediaminetetraacetic acid (EDTA) tubes (BD Diagnostics, USA). Blood was pooled and mixed 1:1 with 1% methylcellulose prepared in RPMI-1640. The mix was centrifuged at 25 x *g* for 15 min at 4˚C. The supernatant was mixed 1:1 with Ca2+- and Mg2+-free Hanks balanced salt solution and layered over the discontinuous Histopaque gradient of 10 ml 1.077 under layering 15 ml of 1.119 and centrifuged for 1 h at 250 x *g* at RT. Heterophils suspended in the 1.119 gradient phase were collected and washed with RPMI-1640 at 425 x *g* for 15 min at 4˚C. Cells were counted, diluted to 1 x $10^7$ cells/ml and stimulated with 10 μg/ml of LPS for 1 h. RNA from BMDCs, BMDMs and heterophils ± LPS was isolated using Qiagen RNeasy kit with on-column DNase digestion according to manufacturer's instructions.

## RNA sequencing and analysis

Total RNA extracted from BMDCs, BMDMs and heterophils (± LPS) was diluted to 100 ng/μl in 20 μl of RNase-free water. The sample preparation was performed by Edinburgh Genomics facility (Roslin Institute, Midlothian, UK) using a Tru-Seq total RNA Sample Preparation v2 kit as per the manufacturer's protocol. Resulting libraries were quality-checked on an Agilent DNA 1000 Bioanalyzer (Agilent Technologies, South Queensferry, UK) and then clustered onto a paired end flowcell using the Illumina TruSeq® Rapid PE Cluster Kit at a 8 pM concentration. The paired-end sequencing, consisting of 100 cycles, was carried out on the Illumina HiSeq 2500 using an Illumina TruSeq® Rapid SBS Kit (Illumina, Little Chesterford, UK). The raw reads were subject to quality control measures, including the removal of remaining sequence adapters. The cleaned, paired-end 100 bp reads were aligned to the chicken reference genome (Galgal4) assembly from the Ensembl database (http://ensembl.org) with TopHat (v2.0.9) splice junction mapper, which aligned reads using Bowtie aligner (v1.0.0). Cufflinks software (v2.1.1) assembled reads into transcripts that were used as input data together with aligned reads in Cuffdiff to determine expression levels by calculating the Fragments per Kilobase per Million mapped reads (FPKM) and the differential expression between conditions using default options. The lists of differentially expressed genes (log2 fold change ≥ 1.0) from the three cell types were compared to each other and genes that were upregulated in two different cell types were added to the list of genes of interest. In addition, the RNA-seq lists were compared to the published studies and additional genes that were upregulated in at least two studies were selected as the genes of interest. RNA-seq data are publicly available via European Bioinformatics Institute ArrayExpress using accession number EMTAB-2996.

## Primer design for cloning of target genes

Candidate gene sequences predicted to reflect transcripts were collected from both Ensembl and NCBI databases. Primers were designed using full-length and/or common regions if there were differences in the predicted sequences between databases or transcripts. If needed, more than one primer pair was designed and tested. For longer cDNAs, partial sequences were used to design primers. To clone full or partial sequences, a panel of cDNA was generated by reverse transcription PCR (RT-PCR) of separate RNA samples isolated from caecal tonsils and spleen from a *Campylobacter* trial [8], bursa from an IBDV trial [9], spleen from IBDV and MDV trial [10], HD11 cells stimulated with LPS, heterophils stimulated with *Salmonella enterica* serovar Enteritidis, or BMDMs, BMDCs and heterophils stimulated with LPS. For each reaction, the following components were added: 1X PCR buffer (-Mg), 0.2 mM of each dNTP, 1.5 mM of MgCl2, 0.5 μM of primers, 1U of Taq DNA Polymerase and 100 ng of template cDNA. All PCRs were performed using an MJ Thermal Cycler (MJ Research). The amplification products were visualised on 1–2% agarose gels and the products of the correct size were excised, purified using a Gel Elution kit (Qiagen) and ligated into the pGEM-T Easy plasmid (Promega Corporation). *E. coli* JM109 competent cells (Promega Corporation) were transformed with the ligation mixtures following the manufacturer's instructions. Colony PCR confirmed the presence of inserts and DNA from selected colonies was extracted and sequenced.

## Primer design for qPCR

All available sequences for a given gene (including the clones above) where compared and primers were designed based on common fragments. All primer designs were performed with Primer Express 3.0 (Applied Biosystems) with consideration of general guidelines. The melting temperature of all primers was set between 58°C and 60°C. The default length of primers was no longer than 30 nucleotides with a GC content in the range of 30–80% where amplicon length did not exceed 150 bp. The last five nucleotides at the 3' end of each primer consisted of ≤ 3 guanines or cytosines and preferably no triplicates of the same base. At least one of the primers overlapped a predicted intron-exon boundary, where possible, to increase specificity of reactions to cDNA derived from transcripts. The primer pairs were tested in qPCR reactions with serial dilutions of pooled cDNA obtained from tissues and cells from the various infection studies specified in the primer design section. A melting curve step was performed to evaluate the specificity of primer pairs. In addition, qPCR products were visualised on agarose gels to confirm the size of the amplicons. Primer design was considered successful once the efficiency of the reaction was between 90% - 105%, with no amplification of genomic DNA and no amplification in No Template Controls (NTCs).

## Sample preparation for high-throughput qPCR

Tissues and blood samples were obtained from eight commercial broilers from the same hatch. Four birds were raised in a high-biosecurity environment (the pedigree farm), and four birds were raised in a farm where the environment resembles broader commercial conditions (sibling-test farm). The pedigree farm birds were vaccinated at the hatchery against MDV and IBV then against coccidiosis at day 5, against Avian Rhinotracheitis (TRT) at day 11 and against IBDV at day 15. Birds housed at the sibling-test farm were vaccinated against TRT, NDV and IBV at hatch and only received IBDV vaccine at day 19. Eights birds were humanely culled at three weeks of age by cervical dislocation. Samples (0.5 x 0.5 cm) of four tissues (bursa, spleen, caecal tonsils and ileum) were collected from the same locations in each bird and stored in RNA*later* until further use. Peripheral blood was collected into tubes containing 5mM ethylenediaminetetraacetic acid (EDTA) and PBLs were isolated on the same day

according to the following protocol. Whole blood was combined with 1% methylcellulose (1:1 ratio) and centrifuged at 25 x $g$ for 15 min at 4°C. The supernatant was carefully collected and mixed with PBS. Blood diluted with PBS was overlaid onto a Histopaque 1.077/1.119 (Sigma Aldrich) discontinuous gradient, prepared by underlying 4 ml of Histopaque 1.077 with 4 ml of Histopaque 1.119 in 15 ml Falcon tubes. The gradient mixture was centrifuged at 400 x $g$ with the brakes off for 30 min at room temperature. Cells were removed from plasma/1.077 Histopaque interface (mononuclear cells) and from the 1.077/1.119 Histopaque interface (heterophils). Cells were combined and washed twice with an equal volume of PBS by centrifugation at 250 x $g$ for 10 min at room temperature. Cells were counted and $10^7$ cells/ml was pelleted and lysed with buffer RLT with β-mercaptoethanol (β-ME) for further total RNA extraction.

## Isolation of RNA and reverse transcription

The RNA*later* stabilised tissues were removed from the reagent using sterile forceps. FastPrepTM Lysing Matrix (MP Biomedicals, USA) tubes containing ~30 mg of tissue were filled with 600 μl of RLT buffer with β-ME. A FastPrep® FP120 Cell Disrupter was used to disrupt and homogenise tissues for 45 sec at a speed of 6.5 m/sec. The lysate was centrifuged for 3 min at 16,000 x $g$ to remove any remaining insoluble material. Total RNA from chicken tissues and PBLs was extracted using an RNeasy Mini Kit (Qiagen) according to manufacturer's instructions. Reverse transcription was performed using the High Capacity Reverse Transcription Kit (Applied Biosystems) according to manufacturer's instructions with random hexamers and oligo (dT)$_{18}$ in a final volume of 10 μl, containing 500 ng total RNA, in G-STORM GS-1 thermal cycler (Gene Technologies). The cDNA samples were stored in -20°C until further use.

## Preamplification

Preamplification of cDNA was performed using TaqMan PreAmp Master Mix (Applied Biosystems). A stock of 200 nM primer mix was prepared by combining an equal concentration of all primers used in the following qPCR. TaqMan PreAmp Master Mix (10 μl) was mixed with 5 μl of 200 nM stock primer mix and 5 μl of diluted cDNA (1:7) in concentration of 185 ng/μl. Samples were incubated at 95°C for 10 min followed by 14 cycles of 95°C for 15 sec and 60°C for 4 min. A clean-up step using Exonuclease I (*E. coli*) (New England Biolabs) was performed to remove unincorporated primers from preamplified cDNA. Exonuclease I was diluted to 4 U/μl and for each 5 μl of preamplified cDNA a total volume of 2 μl Exo I reaction solution was added and incubated at 37°C for 30 min. The reaction was stopped by heating at 80°C for 15 min. Products of preamplification were stored at -20°C.

## High-throughput qPCR using the 96.96 Dynamic Array

Quantitative PCR was performed in the BioMark HD instrument and the 96.96 Dynamic Array (Fluidigm). Assay mixes were prepared by mixing 2.5 μl 2X Assay Loading Reagent (Fluidigm), 2.3 μl of primer pair mix (final concentration 1.15 μM) and 0.2 μl low EDTA TE buffer. Sample mixes were prepared by mixing 2.5 μl TaqMan Gene Expression Master Mix (Applied Biosystems), 0.25 μl 20X DNA Binding Dye Sample Loading Reagent (Fluidigm), 0.25 μl 20X EvaGreen DNA binding dye (Biotum) and 2 μl of preamplified cDNA. Thermal cycling conditions for qPCR were: thermal mix 50°C for 2 min, 70°C for 30 min, 25°C for 10 min, followed by hot start 50°C for 2 min, 95°C for 10 min, PCR (x30 cycles) 95°C for 15 sec, 60°C for 60 sec and melting curve analysis 60°C for 3 sec to 95°C. Real-Time PCR Analysis software 3.1.3 (Fluidigm) was used to visualise results. Analysis settings were as follows: quality

threshold was set to 0.65, baseline correction to linear (derivative) and quantitation cycle (Cq) threshold method to auto (global).

## Validation using conventional quantitative PCR

The conventional 96-well plate format qPCR Applied Biosystems 7500 Fast Real-Time PCR System was used to validate the results from 96.96 Dynamic Array. Three genes were selected based on different patterns of expression when mRNA was compared for all tissues between broilers on the pedigree and sibling-test farms: TNFAIP3 (significantly downregulated), IRG1 (significantly upregulated) and SAAL1 (not affected). The reaction mix was prepared using the following components for each of the samples: 10 µl ABI TaqMan Gene Expression Master Mix (Applied Biosystems), 1 µl 20X EvaGreen (Biotum, VWR-Bie & Berntsen), 2.3 µl 20 µM specific primer pair (forward and reverse) and 4.7 µl nuclease-free water. Each reaction contained 2 µl of cDNA diluted 1:5 in nuclease-free water. The following cycle parameters were used: 2 min at 50˚C, 10 min at 95˚C, followed by 40 or 30 cycles with denaturing for 15 sec at 95˚C and by annealing/elongation for 1 min at 60˚C. Melting curves were generated after each run (from 60˚C to 95˚C, increasing 1˚C/3 sec). The raw Cq data were transformed to log2 values after normalisation based on the most stable reference genes. The Mann-Whitney U test was used for statistical analysis.

## Data and statistical analysis

PCR data pre-processing, normalisation, relative quantification and statistics were performed in GenEx5 and GenEx Enterprise (MultiD Analyses AB). Data were corrected for reaction efficiency for each primer assay individually. The most stably expressed reference genes: TATA box binding protein (TBP), beta-actin (ACTB), glyceraldehyde-3-phosphate dehydrogenase (GAPDH) for the combined tissue dataset, were identified from a panel of seven reference genes using geNorm and NormFinder, as previously described [11]. The geometric means of the most stably expressed reference genes were used to normalise all samples in GenEx5. The normalised dataset repeats were averaged and further normalisation to highest Cq value for a given gene was performed. Relative quantities were transformed to logarithmic scale (log2) before statistical analysis–t-test (GenEx5) and principal component analysis (PCA, XLSTAT).

## Availability of data and materials

The RNA-seq dataset generated and analysed during the current study is available in the European Bioinformatics Institute ArrayExpress using accession number E-MTAB-2996 https://www.ebi.ac.uk/arrayexpress/experiments/E-MTAB-2996/.

Sequences obtained for cloned genes are presented in S1 File.

## Results

### Selection of immune gene panel and sequence confirmation

Transcriptome data from 16 chicken infection studies and 13 studies on mammalian species were used to select upregulated genes (S1 Table) [12–40]. When significantly upregulated genes were compared to gene lists from PubMed search queries of 'innate immune response' and 'gene expression infection' in chickens, humans and other mammalian species, a list of 32 chicken immune-related genes was generated, where significantly upregulated genes were detected in two or more separate studies. To increase the number of the genes of interest, we performed RNA-seq analysis using chicken bone marrow-derived macrophages (BMDMs), bone marrow-derived dendritic cells (BMDCs) and heterophils stimulated with

lipopolysaccharide (LPS) and differentially transcribed genes were identified relative to unstimulated controls and compared across the cell types. Out of 50 (BMDCs) and 69 (BMDMs) differentially transcribed genes annotated in the genome, 24 were common for both cell types. Out of 35 differentially transcribed genes in heterophils, four were common with BMDCs and five with BMDMs. Only one gene, TGM4, was common for all cell types used in analysis (S2 Table). By combining the RNA-seq results with published datasets a final list consisting of 104 candidate genes was compiled, where 32 were selected based on published studies, 24 were selected based on comparison of RNA-seq analysis between BMDMs, BMDCs and heterophils and 48 were selected based on RNA-seq analysis compared with published studies. Out of 104 candidate genes, sequences of 89 genes were confirmed by cloning and sequencing the cognate cDNAs and these were used in primer design for qPCR. Attempts to clone 15 genes were unsuccessful, therefore those genes were removed from the list.

## Validation of the specificity of amplification of target transcripts

To design primer pairs for high-throughput qPCR, the fragments of mRNA sequences that were common for all the transcripts publicly available and in our cloned sequences for a given gene were used. Primer pairs were tested in conventional qPCR using pooled cDNA samples in serial dilutions from various infection studies. The efficiencies of reactions were performed for each pair together with analysis of melting curve that indicated highly specific PCR. Examples are shown in Fig 1. These approaches resulted in universal primer pairs being optimised for each of the 89 genes (S3 Table). In addition, primers for seven reference genes (S4 Table) were optimised in the same manner and used in reference gene normalisation tests using NormFinder and geNorm, as previously described [11].

## Optimisation of reverse transcription, preamplification and primer pairs

According to the manufacturer's instructions the PreAmp Master Mix was optimised on total RNA reverse transcribed using a High Capacity Reverse Transcription (RT) Kit. The preamplification reaction was optimised in 20 μl reaction containing185 ng/μl of cDNA with 14 cycles of PCR.

The 96.96 Dynamic Array was used to test whether limiting dilutions of cDNA generated from standard RNA are detectable in the 96.96 Dynamic Array. The RNA was extracted from COS-7 cells transfected with plasmids containing sequences for IL1B, IL6, IL12B, IL18 and CXCLi2, as previously described [41]. As shown in Fig 2A the array detected cDNA derived from these transcripts in all dilutions and the Cq values increased with decreasing concentration of cDNA. The array was also validated using RNA from chicken tissues that were reverse transcribed and preamplified. The non-preamplified cDNA samples were used in comparison. As shown in Fig 2B the preamplified samples have lower Cq values compared to the corresponding cDNA that was not preamplified. The tests confirmed that the reverse transcription, preamplification and primer pairs work efficiently with 96.96 Dynamic Array.

## Gene expression analysis using the 96.96 Dynamic Array

To explore the potential utility of the 96.96 Dynamic Array in quantifying innate immune gene expression, RNA extracted from tissues from eight broilers raised in varying biosecurity environments (pedigree and sibling-test farms) was analysed. The experiment sought to determine if the assays are sensitive enough to detect expression of innate immune gene transcripts in nanoliter qPCRs. The samples and assays were loaded separately in 5 μl volumes into the wells of the microfluidic array and redistributed into the 9,216 chambers followed by mixing of assays and preamplified cDNA. Successful runs were confirmed by checking images of the

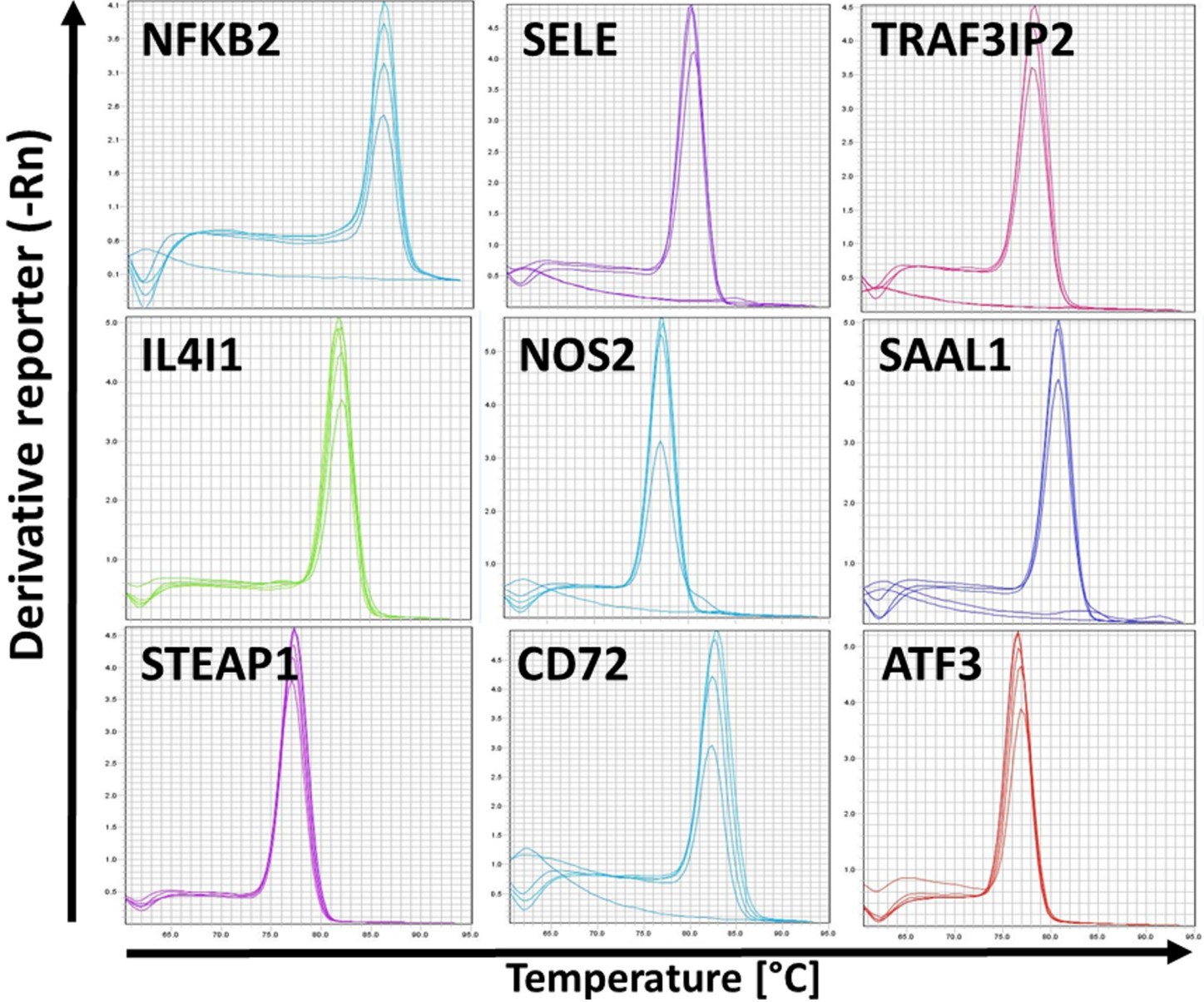

**Fig 1. Optimisation of primer pairs.** Examples of melting curve analysis based on serial dilutions of pooled cDNA generated from tissues/cells isolated from various infection studies.

loaded chip (ROX signal) and the heat map view generated by the instrument (Fig 3). The duplicate samples had similar Cq values and the overall difference between duplicates Cq was no greater than 0.5, which confirms reliability of the reactions. The volume of reaction in each of the chambers was 6 nl and for low expressed genes fluorescence was detected in only one chamber and for others was not detected at all, as represented by a black filled square. These were removed from the analysis in GenEx software. Most of the non-template control chambers did not contain a fluorescence signal, apart from ENSGALG00000015395 gene with Cq value > 28.5. The Cq values were then transformed into the log2 values in GenEx software for further analysis.

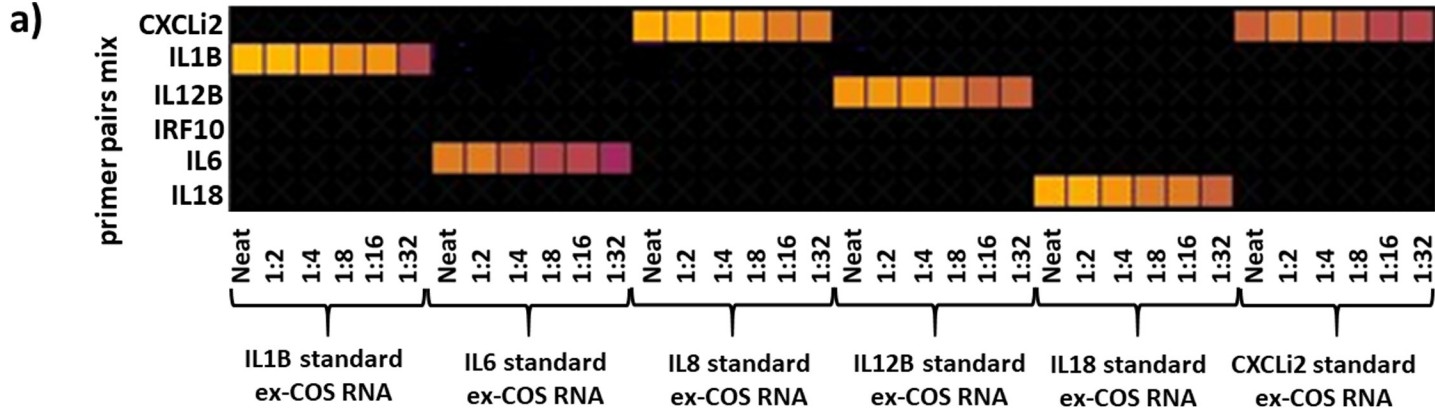

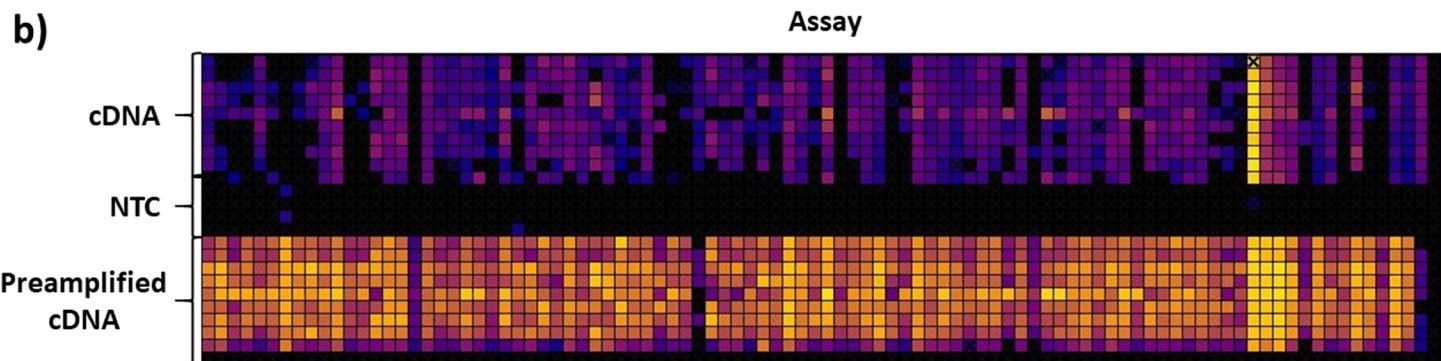

**Fig 2. Validation of 96.96 Dynamic Array.** A) standard RNA samples of COS-7 cells transfected with plasmids containing genes for: CXCLi2, IL1B, IL12B, IL6, IL18, reverse transcribed and used in limiting dilutions; B) cDNA and cDNA after preamplification showing higher amplification in the later; brighter colour indicates lower Cq value, higher amplification.

The expression of the 89 genes selected for this study in 40 samples was analysed, representing five tissues from each of 4 birds separately reared on the two test farms. When comparing the whole dataset by Principal Component Analysis (PCA) four groups were identified (Fig 4). The differences in expression profiles between farms were not visible based on the spleen, bursa, caecal tonsils and ileum samples. The organ tissues were clustered together with no clear sets consisting of tissues from birds housed on a particular farm. However, the PBL samples from pedigree and sibling-test farm clearly separated by PCA, indicating differences in gene expression profiles between PBLs from these two groups. In addition, PCA was performed on the same dataset with PBL samples removed. This second analysis showed heterogeneity within the tissues where bursa and spleen samples grouped together and gut derived samples (caecal tonsil and ileum), slightly overlapped, but there was no separation based on the farm type (Fig 5). The analysis of gene expression levels between farms where all tissues were taken under consideration showed only 13 genes to be differentially regulated. To explore the difference between farms, individual tissues were analysed and the number of significantly differentially transcribed genes for each tissue were: 19 for bursa, 12 for spleen, 9 for caecal tonsils and 23 for ileum (Fig 6 and Table 1). There was a clear direction of gene regulation in bursa, where most of the genes were downregulated in samples collected from sibling-test farm. In contrast, in ileum and PBL samples the majority of genes were upregulated. Spleen and caecal tonsils had similar numbers of genes up- and downregulated. The number of significantly differentially transcribed genes between farms in PBL samples was much higher

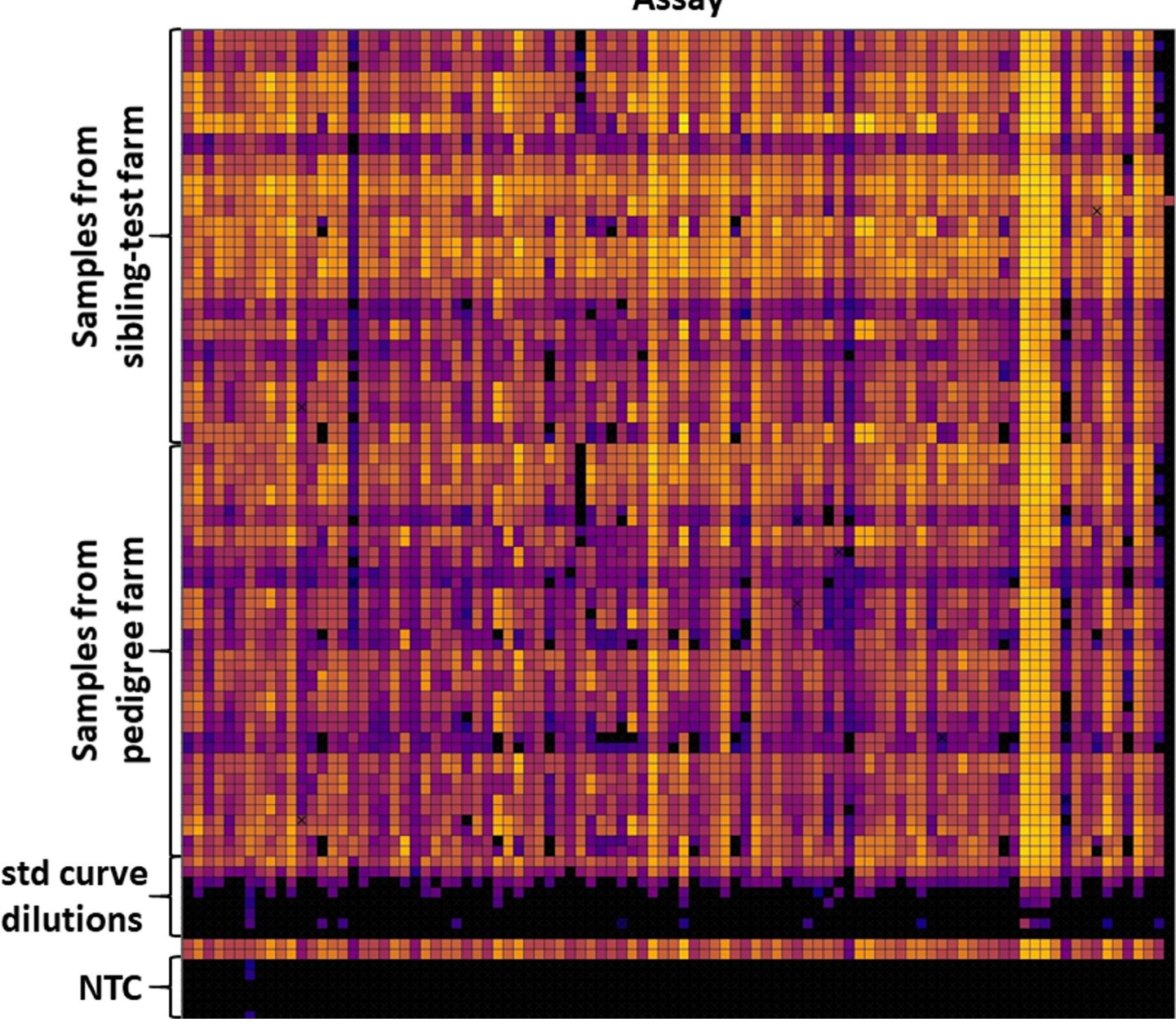

**Fig 3. Heat map view of the final 96.96 Dynamic Array run with individual assays on the x axis and individual samples on y axis.** The RNA extracted from tissue samples from broilers raised in two different biosecurity environments (sibling-test and pedigree farms); samples from five tissues from four birds on each of the two farms were used; pooled preamplified cDNA was used for standard curve dilutions.

compared to the tissues samples where 51 significantly differentially transcribed genes were found with only 5 genes downregulated. The fold change values for all tissues analysed together and for separate comparison of tissues between farms are presented in S5 Table.

## Validation of differentially expressed genes by conventional qPCR

To validate the 96.96 Dynamic Array qPCR results, three transcripts (TNFAIP3, IRG1, SAAL1) were analysed by qPCR in a conventional 96-well format. These genes were selected

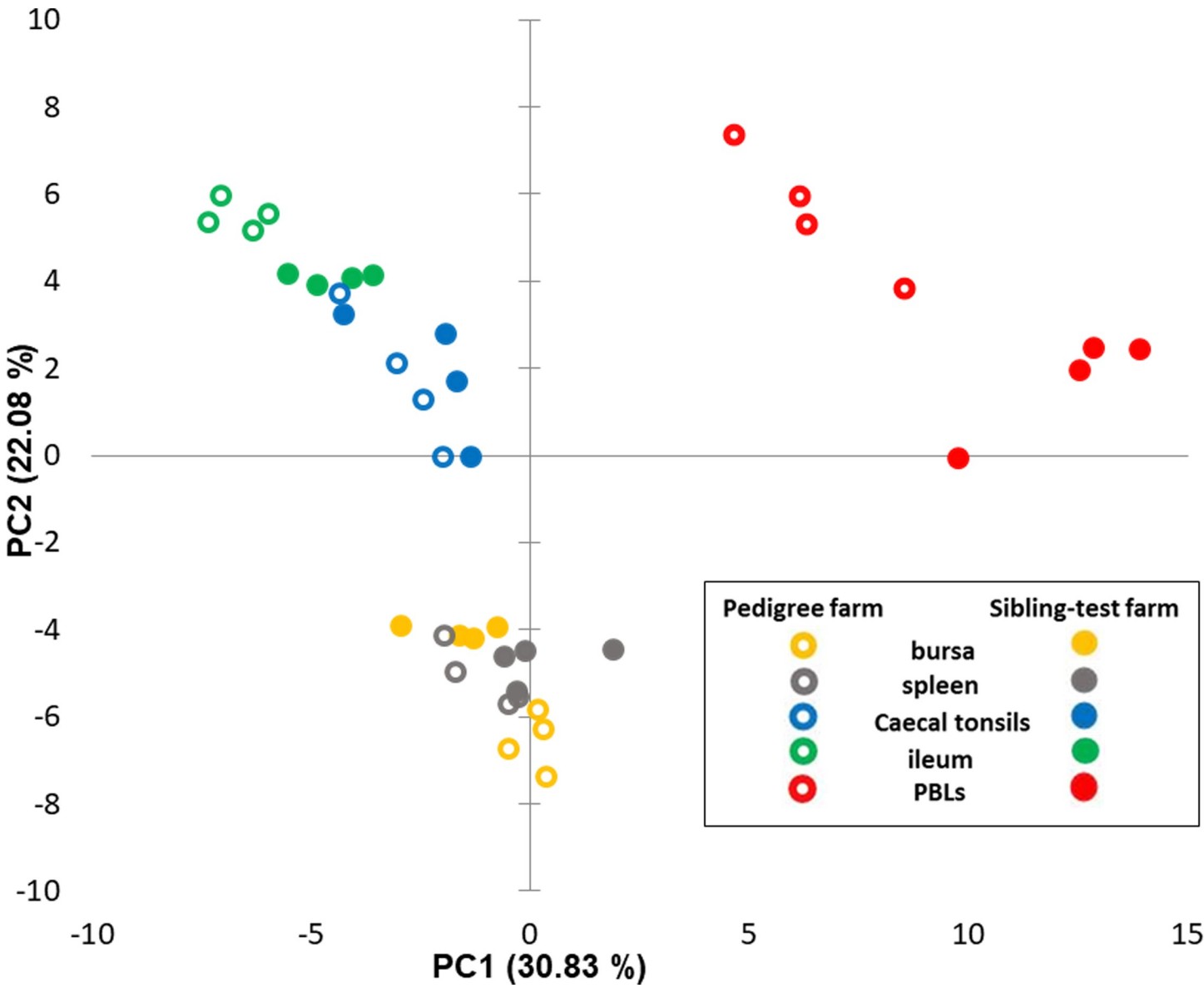

**Fig 4. Principal component analysis for 96.96 Dynamic Array including all tissues datasets.** Analysis indicates broad similarities and differences in transcription of immune-related genes in bursa, spleen, ileum and caecal tonsils and peripheral blood leukocytes (PBLs); data points represent individual samples of sibling-test (filled markers) and pedigree farms (no fill markers).

because they exhibited consistent but distinct patterns of regulation in all tissues from both farms, being either downregulated (TNFAIP3), upregulated (IRG1) and showing no change in expression (SAAL1) compared to control samples from the pedigree farm. The qPCR data for TNFAIP3, IRG1 and SAAL1 across the test farms by 96.96 Dynamic Array and conventional qPCR are shown in Fig 7. For TNFAIP3, the same trend of down-regulation was detected, for IRG1 a significant upregulation was detected, and for SAAL1 no change in expression was detected by conventional qPCR as observed using the 96.96 Dynamic Array.

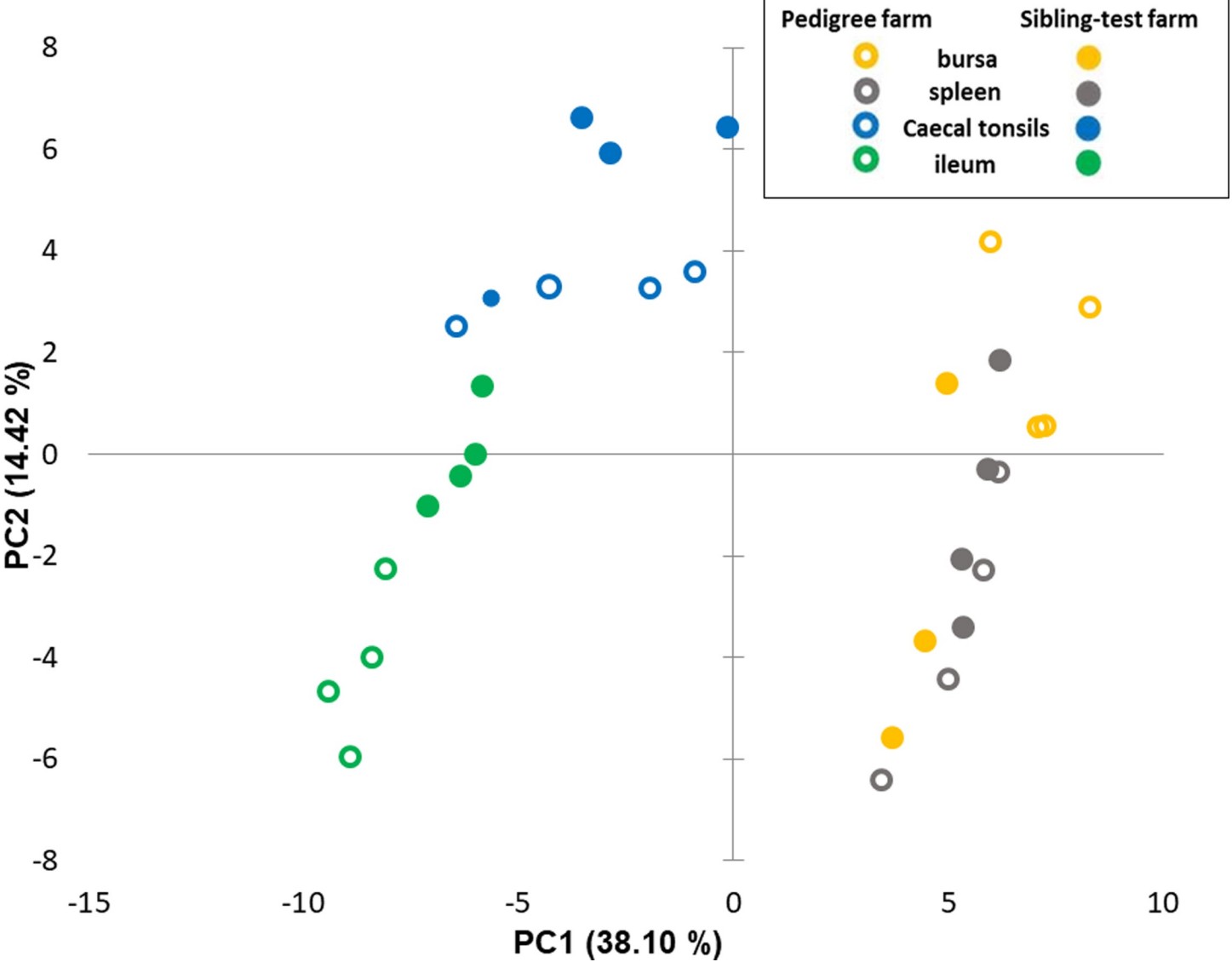

**Fig 5. Principal component analysis for 96.96 Dynamic Array excluding PBLs dataset.** Analysis of immune-related genes in bursa, spleen, ileum and caecal tonsils of birds reared on the sibling test (filled markers) and pedigree (no fill markers) indicates heterogeneity within the tissues and no separation based on the farm type.

## Discussion

Here, we report the development, validation and application of a high-throughput qPCR-based tool for assessment of chicken immune gene expression. The 96.96 Dynamic Array within the BioMark system allows simultaneously analysis of 96 biological samples for transcription of 96 genes of interest, generating Cq values from 9,216 reactions in one run. It was applied here to 89 chicken immune-related genes and panel of reference genes, which we selected on the basis of differential transcription in response to pathogens or their constituents. All the selected genes were confirmed by cloning and sequencing. Specificity of the primer pairs was confirmed by melting curve analysis and agarose gel electrophoresis of amplicons. The experiments reported here were successful but running the array has its challenges. The reaction chambers will not be loaded if there is an air bubble in the well, which may result in

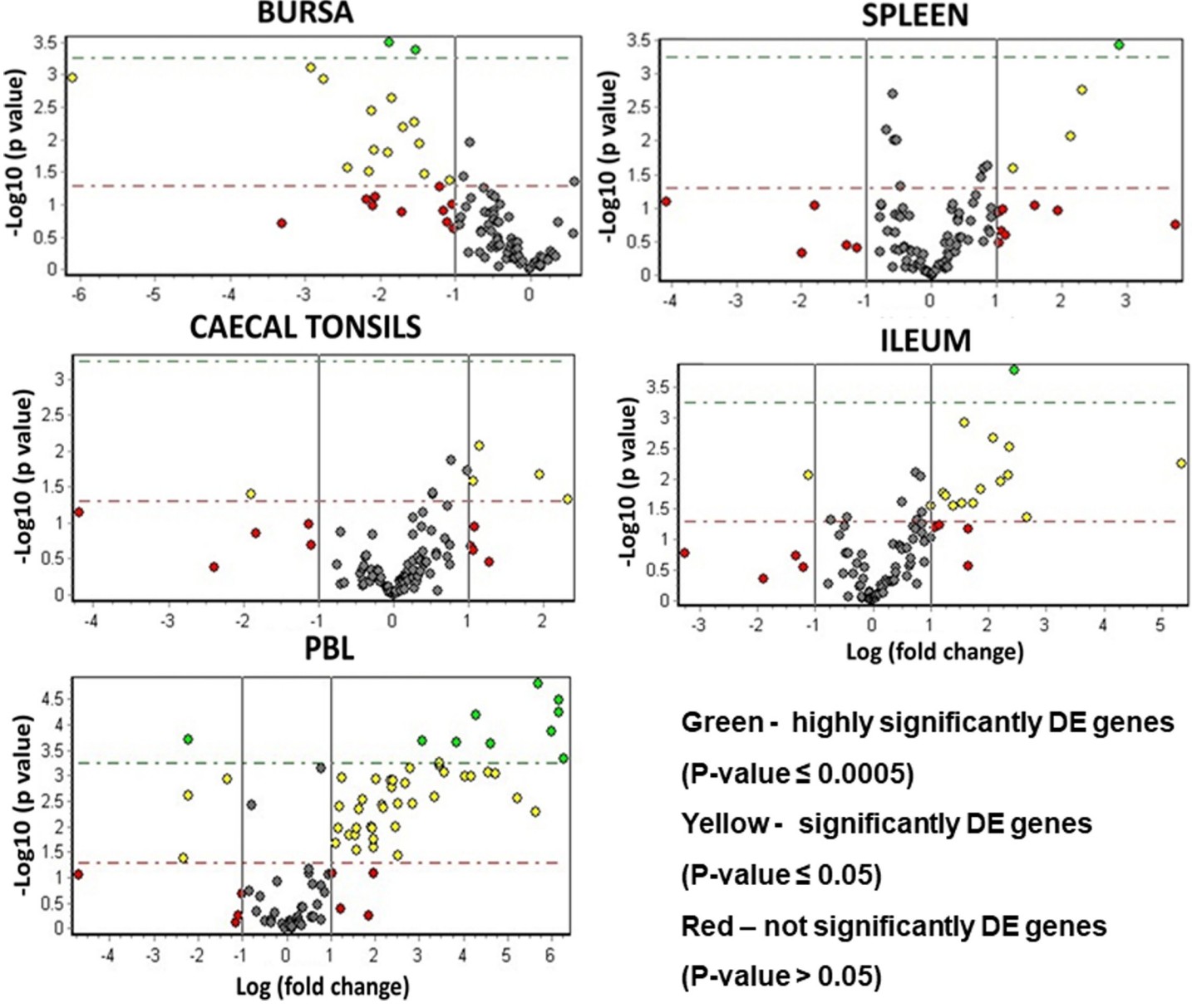

**Fig 6. Volcano plots of immune-related genes expressed in tissues collected from chickens raised on pedigree and sibling test farms.** Scattered points represent genes; the x-axis is the log2 fold change for the ratio sibling test vs pedigree farm, whereas y-axis is the log10 p-value.

multiple reactions not being performed. To avoid that, there is a need for careful pipetting and making sure that all the air bubbles are removed from wells before the sample and assay mixes are distributed into the chambers. Although the array priming, loading and qPCR are not time-consuming, the optimisation of sample preparation is. It proved to be crucial that the reverse transcription and preamplification steps are validated before the samples are used in the 96.96 Dynamic Array.

For the purpose of this study, the selection criteria aimed to define avian genes that play important roles in immune responses to various pathogens or constituents thereof. The newly developed tool could have broader applications in studies on general immune responsiveness to diseases and/or vaccines. Detection of differences in immune performance were performed

**Table 1. Lists of genes differentially expressed in individual tissues.**

| Sibling test farm[a] vs pedigree farm[b] | | | | |
|---|---|---|---|---|
| **All tissues** | **Bursa** | **Spleen** | **Caecal tonsils** | **Ileum** |
| DTX2 ↓ | MAFF ↓ | IRG1 ↑ | IL4I1 ↑ | ENSGAL27419 ↑ |
| IRG1 ↑ | CSF1 ↓ | HPS5 ↑ | BATF3 ↑ | NOS2 ↑ |
| ENSGAL15395 ↓ | ENSGAL22324 ↓ | SERPINE2 ↓ | ENSGAL22324 ↑ | CXCLI1 ↑ |
| IL1R2 ↑ | PKD2L1 ↓ | NFKBIZ ↓ | HPS5 ↑ | HPS5 ↑ |
| SNX10 ↑ | G0S2 ↓ | IL4I1 ↑ | IL13RA2 ↑ | PKD2L1 ↑ |
| IL4I1 ↑ | ENSGAL5747 ↓ | CSF1 ↓ | PPARG ↑ | PLK3 ↑ |
| ENSGAL27419 ↑ | TLR15 ↓ | ENSGAL11172 ↓ | ENSGAL15395↓ | IRG1 ↑ |
| IL13RA2 ↑ | STEAP1 ↓ | EAF2 ↑ | SNX10 ↑ | EDN1 ↓ |
| IRF10 ↑ | SERPINE2 ↓ | SOCS3 ↑ | PKD2L1 ↑ | SNX10 ↑ |
| HPS5 ↑ | SOCS3 ↓ | SLCO6A1 ↑ | | TLR15 ↑ |
| TNFAIP3 ↓ | IL6 ↓ | GCH1 ↑ | | CXORF21 ↑ |
| IL18↑ | RASD1 ↓ | NOS2 ↓ | | CCL20 ↑ |
| CCLI3 ↑ | SLCO6A1 ↓ | | | RASD1 ↓ |
| | MADPRT ↓ | | | ENSGAL5747 ↑ |
| | TNFAIP3 ↓ | | | IL4I1 ↑ |
| | ABCG2 ↓ | | | CXCL13L2 ↑ |
| | CXORF21 ↓ | | | CD72 ↑ |
| | NR4A3 ↓ | | | ENSGAL22324 ↑ |
| | CCL19 ↑ | | | SOCS3 ↑ |
| | | | | LYG2 ↑ |
| | | | | CSF1 ↓ |
| | | | | IRF7 ↑ |
| | | | | GLUL ↓ |
| **Peripheral blood leukocytes** | | | | |
| | IRG1 ↑ | CD83 ↑ | CSF1 ↑ | |
| | IL4I1 ↑ | CXCLI2 ↑ | IL13RA2 ↑ | |
| | PTGS2 ↑ | EGR1 ↑ | IL1B ↑ | |
| | ATF3 ↑ | LYZ ↑ | PLK3 ↑ | |
| | TGM4 ↑ | ENSGAL25905 ↑ | ABCG2 ↑ | |
| | BATF3 ↓ | IL1R2 ↑ | TLR4 ↑ | |
| | CXORF21 ↑ | TNIP2 ↑ | EAF2 ↑ | |
| | CCLI4 ↑ | ENSGAL22324 ↑ | MAFF ↑ | |
| | ENSGAL27419 ↑ | IL10RA ↑ | RASD1 ↑ | |
| | IL6 ↑ | GCH1 ↓ | C3ORF52 ↑ | |
| | PPARG ↑ | BCL2A1 ↑ | SDC4 ↑ | |
| | HPS5 ↑ | CCL20 ↑ | IL12B ↑ | |
| | SOCS3 ↑ | NFKBIZ ↑ | CD40 ↑ | |
| | SNX10 ↑ | TLR15 ↑ | EDN1 ↑ | |
| | CXCL13L2 ↓ | UPP1 ↑ | G0S2 ↑ | |
| | ETS2 ↑ | SELE ↓ | SLCO6A1 ↑ | |
| | STEAP4 ↑ | IL18 ↑ | ENSGAL15395 ↓ | |

Legend: ↑ - upregulation; ↓- downregulation

[a] sibling-test farm samples (n = 4)

[b] pedigree farm samples (n = 4).

to establish which tissue delivers the most informative data. The gene expression was anticipated to differ due to dissimilar biosecure environments on the farms. The difference in gene expression profiles were most obvious among PBL samples. The relative gene expression analysis of PBLs showed that differentially transcribed genes were not only of proinflammatory functions, but additionally significant expression of genes involved in diminishing inflammation. The PBLs consist of many cells programmed to control microorganisms as the first line of innate defence and showed to be promising indicator of immune gene expression differences between farms. Additional validation experiments were performed where selected RNA samples of both farms were again transformed into preamplified cDNA and used in

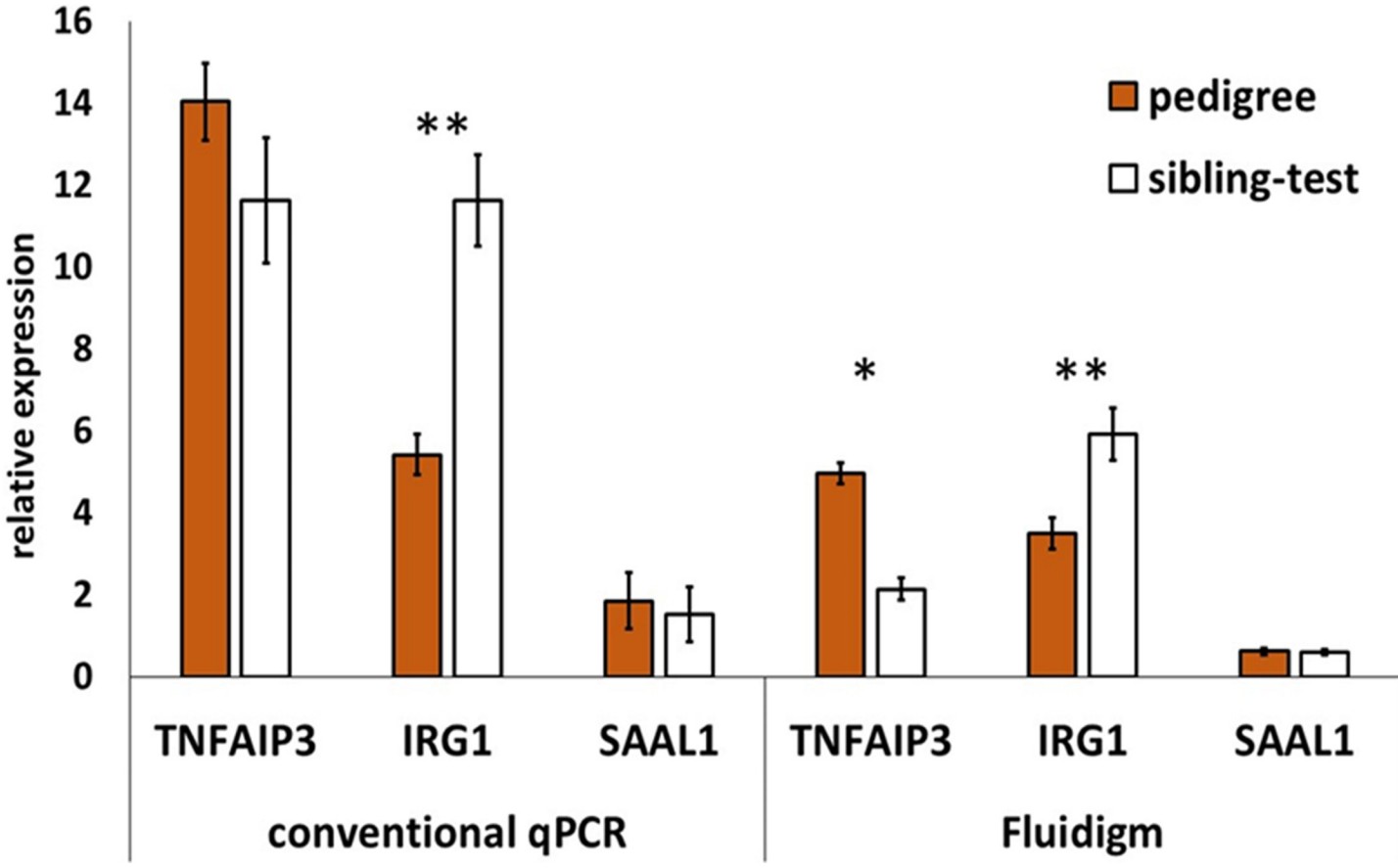

**Fig 7. Validation of 96.96 Dynamic Array qPCR results for selected 3 transcripts: TNFAIP3, SAAL1, and IRG1.** Relative expression of three genes tested between sibling-test and pedigree farm in 96.96 Dynamic Array, and in conventional 96-well plate format qPCR Applied Biosystems 7500 Fast Real-Time PCR System; each bar represent average of biological replicates from 4 birds and 5 tissues; error bars depict SEM, **—P-value ≤ 0.0005; *—P-value ≤ 0.05.

quadruplicates in 96.96 Dynamic Array to test the reproducibility of all steps involved. The experiments reported here were successful and based on Cq values generated similar datasets (S1 Fig).

The poultry industry must overcome many challenges caused by diseases that lead to decreased production and bird welfare. Implementation of selection goals for improved innate immune responses in commercial breeding programmes could play a significant role in minimising these production losses. Current commercial strategies to achieve this goal rely on multi-environment selection strategies, where performance estimates on relatives of selection candidates housed in non-bio-secure environments are used to inform selection decisions within a breeding programme [42]. The availability of a more specific method of evaluating chickens with better resilience or immune responsiveness to pathogens could support greater progress. As a cost-effective and rapid technique to support the selection of birds, the array could potentially be used as a screening aid particularly to describe immune status for example during the transitional period prior to significant microbial challenge, early after hatch or even during late stages of embryo development. Mortality in the first week post-hatch is an important aspect of chicken breeding and is used as an indicator of the occurrence of infection. The gene expression screening of newly-hatched chicks, yet to have significant contact with pathogens or acquired microbiota could help to predict how the flock will perform in later stages. In

the present study, where immune-related gene expression was analysed in 3-week-old birds on farms of varying biosecurity, the differences detected in PBL gene expression profiles reflect varying levels of exposure to pathogens, rather than the basal immune status of the birds sampled.

The value of the multiplex PCR platform for immune-related genes could also be tested in a wider range of environmental settings, for example on test farms using birds of the same genotype but reared in cold and hot environments, at different stocking densities, intensive indoor vs. outdoor free-range systems and so on. Expansion of breeding under hot temperatures is important because of global climate change. Heat stress is known to result in decreased productivity and increased mortality [43]. Several heat shock proteins and genes responsible for glucose transportation have been shown to be involved in responses to heat stress [44–46]. These, and many other expression quantitative trait loci (eQTL) associated with heritable resistance to other production-relevant phenotypes, could be added to the qPCR panel to test differences in transcript expression between farms as a guide to selection.

Apart from the application in breeding programmes, the gene panel used within the 96.96 Dynamic Array could be utilised in studies focused in detail on immune responses to particular pathogens. Generally, in those experiments whole transcriptome sequencing is applied as a technique for novel genes detection or conventional qPCR with the focus on only few selected genes. The genes selected for this study were of significance in immune responses to broad spectrum of pathogens in various cells and tissues. Therefore, this platform and carefully selected gene panel are more convenient to study immune responses without the need of costly sequencing of many samples and multiple time points.

The qPCR gene panel was used to test the immune responses to changes in biosecurity levels but it could be further refined by addition of infection-specific genes to analyse responses to particular pathogen. This approach was used by Dalgaard et al., (2015) [47] where a panel of immune genes was tested against samples collected from *Ascaridia galli* infected chickens at different time points of infection, which spanned different phases of worm development. With addition of pathogen-specific genes the qPCR panel could as well be used to establish responsiveness of chickens to vaccination and if this reaction is compromised by secondary infections. Recently a panel of primers was developed for specific detection of the significant respiratory pathogens of poultry and tested in 96.96 Dynamic array chip [48]. The validation of primers designed against avian respiratory pathogens showed that primers do not cross react, and can therefore be used to detect co-infections in avian species.

## Conclusions

Taken together, the carefully selected genes and the rapidity and cost-effectiveness of the analysis using the high-throughput qPCR 96.96 Dynamic Array make this a convenient tool for measuring differential avian immune responses in various settings.

## Supporting information

**S1 Table. Results of search queries for creation of the gene list.**
(XLSX)

**S2 Table. List of differentially expressed genes from RNA-seq analysis.**
(XLSX)

**S3 Table. Primers, alignment site and amplicon length for qPCR detection of transcripts of genes of interest using the Fluidigm 96.96 Dynamic Array in BioMark system.**
(XLSX)

**S4 Table. Reference gene primers and accession numbers.**
(XLSX)

**S5 Table. Fold change in gene expression between sibling-test and pedigree farms.**
(XLSX)

**S1 Fig. Heat map view of the 96.96 Dynamic Array validation experiment.** The individual assays on the x axis and selected individual samples from both farms in quadruplicates on y axis.
(TIF)

**S1 File. Sequence alignments for cloned genes.**
(DOCX)

**S2 File. NC3Rs ARRIVE guidelines checklist.**
(PDF)

## Author Contributions

**Conceptualization:** Kellie A. Watson, Pete Kaiser, Mark P. Stevens.

**Data curation:** Dominika Borowska, Richard Kuo.

**Formal analysis:** Dominika Borowska, Richard Kuo.

**Funding acquisition:** Kellie A. Watson, Pete Kaiser, Lonneke Vervelde, Mark P. Stevens.

**Investigation:** Dominika Borowska, Richard Kuo, Richard A. Bailey, Kellie A. Watson, Pete Kaiser, Lonneke Vervelde, Mark P. Stevens.

**Methodology:** Dominika Borowska, Pete Kaiser, Lonneke Vervelde, Mark P. Stevens.

**Supervision:** Pete Kaiser, Lonneke Vervelde, Mark P. Stevens.

**Visualization:** Dominika Borowska.

**Writing – original draft:** Dominika Borowska.

**Writing – review & editing:** Richard Kuo, Richard A. Bailey, Kellie A. Watson, Lonneke Vervelde, Mark P. Stevens.

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
