## [Decision Letter · Decision Letter 0]

11 Oct 2019

PONE-D-19-22735

Highly multiplexed quantitative PCR-based platform for evaluation of chicken immune responses

PLOS ONE

Dear Dr Borowska,

Thank you for submitting your manuscript to PLOS ONE. After careful consideration, we feel that it has merit but does not fully meet PLOS ONE’s publication criteria as it currently stands. Therefore, we invite you to submit a revised version of the manuscript that addresses the points raised during the review process.

Authors should prepare reasoned comments for Reviewer #1 and make the necessary changes to the manuscript.

We would appreciate receiving your revised manuscript by Nov 25 2019 11:59PM. To enhance the reproducibility of your results, we recommend that if applicable you deposit your laboratory protocols in protocols.io, where a protocol can be assigned its own identifier (DOI) such that it can be cited independently in the future. For instructions see: http://journals.plos.org/plosone/s/submission-guidelines#loc-laboratory-protocols

We look forward to receiving your revised manuscript.

Kind regards,

Ruslan Kalendar, PhD

Academic Editor

PLOS ONE

**Journal Requirements**

**Comments to the Author**

1. Is the manuscript technically sound, and do the data support the conclusions?

Reviewer #1: Yes

Reviewer #2: Yes

2. Has the statistical analysis been performed appropriately and rigorously? 

Reviewer #1: Yes

Reviewer #2: I Don't Know

3. Have the authors made all data underlying the findings in their manuscript fully available?

Reviewer #1: Yes

Reviewer #2: Yes

4. Is the manuscript presented in an intelligible fashion and written in standard English?

Reviewer #1: Yes

Reviewer #2: Yes

5. Review Comments to the Author

Reviewer #1:

This is a well written and justified presentation of a new method for analyzing chicken immune responses where other reagents are lacking. While this new method relies on analysis of RNA expression, this is an adequate first step when bulk protein analysis cannot be conducted due to limited reagents.

Reviewer #2:

The manuscript by Borowska et al., describes the generation of a high-throughput qPCR analysis approach for measuring 89 chicken immune genes. The genes were selected from a series of published RNA seq papers, and differentially expressed genes were identified. RNA samples were made from a series of isolated and challenged tissues, and cell types (bone marrow derived macrophages, dendritic cells, etc. ) PCR products were cloned and sequenced to validate the primer pairs for each gene. Primers spanned an intron where possible. The multiplex PCR was validated by examining the differential gene expression between a breeder farm and a sibling test farm (different biosecurity level). This is an extremely valuable tool for measuring chicken immune responses to pathogens.

While the PCR products were validated individually, it is not clear to me that the same products would only be produced in the PCRs involving the primer mixes, where many more things could potentially interfere. I would encourage the authors to further validate this primer set prior to publication.

I do not understand the rationale for the comparison of the two farms. While this study is interesting and potentially valuable to the industry partners, this would seem to have too many uncontrolled variables to actually be useful to test the 96.96 Fluidigm approach. I would have much preferred to examine the original RNAs used to identify the DE genes (agonist stimulated DC or M). Are these birds genetically related? Are they siblings? Is something known of their exposure to various pathogens. Any information would be helpful.

Fig. 6 One of the three genes in the test by qPCR does shows the same pattern

In Table 1. Gene name is probably RSAD1

6. PLOS authors have the option to publish the peer review history of their article (what does this mean?). If published, this will include your full peer review and any attached files.

Reviewer #1: Yes: Charles T Spencer

Reviewer #2: No

---

## [Author Response · Author response to Decision Letter 0]

7 Nov 2019

Dear Editor and Reviewers,

Thank you for your email and the opportunity to revise our manuscript entitled ‘Highly multiplexed quantitative PCR-based platform for evaluation of chicken immune responses’ (PONE-D-19-22735). We wish to thank the reviewers for their careful reading of this manuscript and for their constructive suggestions and comments. We have revised the manuscript accordingly and consider that the manuscript is improved as a consequence. Our response to the specific comments raised by each reviewer are detailed below, with the reviewer’s comments in italic type font and all references to text line numbers in parentheses are to the revised manuscript. As requested we have used track changes in the manuscript to indicate where the changes have been made and these can be found in the document ‘Revised Manuscript with Track Changes’.

We note that you have included the phrase “data not shown” in your manuscript. Unfortunately, this does not meet our data sharing requirements. PLOS does not permit references to inaccessible data. We require that authors provide all relevant data within the paper, Supporting Information files, or in an acceptable, public repository. Please add a citation to support this phrase or upload the data that corresponds with these findings to a stable repository (such as Figshare or Dryad) and provide and URLs, DOIs, or accession numbers that may be used to access these data. Or, if the data are not a core part of the research being presented in your study, we ask that you remove the phrase that refers to these data.

Where we indicated ‘data not shown’ in the original submission we have now removed the statement or provided additional information in the revised manuscript as follows:

- Line 287 – 288 removed

- Line 291 information added

- Line 299 information removed

- Line 411 supplementary figure added

Reviewer #1:

This is a well written and justified presentation of a new method for analyzing chicken immune responses where other reagents are lacking. While this new method relies on analysis of RNA expression, this is an adequate first step when bulk protein analysis cannot be conducted due to limited reagents.

Thank you for a positive assessment. Your recognition of our work is much appreciated. 

Reviewer #2:

The manuscript by Borowska et al., describes the generation of a high-throughput qPCR analysis approach for measuring 89 chicken immune genes. The genes were selected from a series of published RNA seq papers, and differentially expressed genes were identified. RNA samples were made from a series of isolated and challenged tissues, and cell types (bone marrow derived macrophages, dendritic cells, etc.) PCR products were cloned and sequenced to validate the primer pairs for each gene. Primers spanned an intron where possible. The multiplex PCR was validated by examining the differential gene expression between a breeder farm and a sibling test farm (different biosecurity level). This is an extremely valuable tool for measuring chicken immune responses to pathogens.

While the PCR products were validated individually, it is not clear to me that the same products would only be produced in the PCRs involving the primer mixes, where many more things could potentially interfere. I would encourage the authors to further validate this primer set prior to publication.

We have incorporated multiple steps to ensure that the primer pairs used in this study are most likely to be specific for one gene and not produce multiple products when applied in a mixture. Firstly, the primer pairs used in this study are gene-specific and have been carefully designed following the guidelines, as described in “Primer design for qPCR section”, lines 172 – 178. At least one of the primer (forward or reverse) was designed to span exon-intron boundary, in case of multi-exon genes, as described in the S3 Table.

Secondly, the melting curve analysis was performed on each of the primer pairs in every reaction and only a single peak was detected regardless of the template sample used (cDNA or pre-amplified cDNA). If the primers were to align to non-specific sequences that would be detected in the melting plots as an additional peak or peaks relative to the target amplicon. 

Finally, the cDNA samples were pre-amplified with the primer mixture prior to the use on 96.96 Dynamic Array. If non-specific amplification occurs then this will not be repeated on the array because individual primer pairs are used in the separate chambers, amplifying only one gene. The DNA melting profile of each individual chamber on the array is indicative for its specificity and therefore we know that only specific products were amplified in the reactions.

I do not understand the rationale for the comparison of the two farms. While this study is interesting and potentially valuable to the industry partners, this would seem to have too many uncontrolled variables to actually be useful to test the 96.96 Fluidigm approach. I would have much preferred to examine the original RNAs used to identify the DE genes (agonist stimulated DC or M). Are these birds genetically related? Are they siblings? Is something known of their exposure to various pathogens. Any information would be helpful.

The gene list was designed to fit both purposes, to study primary cell cultures as well as a tool to screen performance of flocks raised under different environments. The decision was made to test the gene panel within the 96.96 Dynamic Array on samples collected from genetically related broilers that were housed in two different environments: a pedigree farm – representing a highly biosecure environment where breeding-programme selection candidates are recorded and selected, and the sibling-test farm - a non-bio-secure environment aimed to resemble broader commercial conditions and where full-siblings and half-siblings of selection candidates are placed. The vaccination schedule is described in lines 187-190. The farms differ greatly in microbial load but the exact composition of microbial community in the litter was not tested during this study.

The bone marrow derived dendritic cells, heterophils and macrophages were derived from J line Brown Leghorn layers. This is an outbred population housed in National Avian Research Facility under conventional farm settings. The vaccination schedule of the J line chicken is described in lines 105-109. Additional pathogens may be present in the litter but were not tested during this study.

Fig. 6 One of the three genes in the test by qPCR does shows the same pattern

The profile of expression detected by qPCR is consistent with that observed using 96.96 Dynamic Array. We agree that only one gene tested in conventional qPCR has a significant difference in expression between farms, however the pattern is the same for all of the genes.

In Table 1. Gene name is probably RSAD1

The gene symbol in the table 1 does not refer to the RSAD1 gene (radical S-adenosyl methionine domain containing 1). The gene symbol is correct - RASD1 ras related dexamethasone induced 1, NM_001044636, ENSGALG00000004860, as described in S3 Table.

We are grateful for your editorial assistance and will be pleased to clarify any aspects. 

Yours sincerely

Dr Dominika Borowska, The Roslin Institute & Royal (Dick) School of Veterinary Studies, University of Edinburgh.

---

## [Editor Report · Decision Letter 1]

11 Nov 2019

Highly multiplexed quantitative PCR-based platform for evaluation of chicken immune responses

PONE-D-19-22735R1

Dear Dr. Borowska,

We are pleased to inform you that your manuscript has been judged scientifically suitable for publication and will be formally accepted for publication once it complies with all outstanding technical requirements.

With kind regards,

Ruslan Kalendar, PhD

Academic Editor

PLOS ONE

---

## [Editor Report · Acceptance letter]

19 Nov 2019

PONE-D-19-22735R1 

Highly multiplexed quantitative PCR-based platform for evaluation of chicken immune responses 

Dear Dr. Borowska:

I am pleased to inform you that your manuscript has been deemed suitable for publication in PLOS ONE. Congratulations! Your manuscript is now with our production department. 

With kind regards,

on behalf of

Dr. Ruslan Kalendar 

Academic Editor

PLOS ONE